# Untargeted lipidomic analysis of metabolic dysfunction-associated steatohepatitis in women with morbid obesity

**Laia Bertran[1], Jordi Capellades[2], Sonia Abelló[3], Cristóbal Richart [1]***

**1** Department of Medicine and Surgery, Rovira i Virgili University, Tarragona, Spain, **2** Department of Electronic, Electric and Automatic Engineering, Higher Technical School of Engineering, Rovira i Virgili University, Tarragona, Spain, **3** Scientific and Technical Service, Rovira i Virgili University, Tarragona, Spain

* crichartjurado@gmail.com

## Abstract

Metabolic Dysfunction-Associated Steatohepatitis (MASH) represents the severe condition of Metabolic Dysfunction-Associated Steatotic Liver Disease (MASLD). Currently, there is a need to identify non-invasive biomarkers for an accurate diagnosis of MASH. Previously, omics studies identified alterations in lipid metabolites involved in MASLD. However, these studies require validation in other cohorts. In this sense, our aim was to perform lipidomics to identify the circulating lipid metabolite profile of MASH. We assessed a liquid chromatography coupled to a mass spectrometer-based untargeted lipidomic assay in serum samples of 216 women with morbid obesity that were stratified according to their hepatic diagnosis into Normal Liver (NL, n = 44), Simple Steatosis (SS, n = 66) and MASH (n = 106). First, we identified a profile of lipid metabolites that are increased in MASLD, composed of ceramides, triacylglycerols (TAG) and some phospholipids. Then, we identified that patients with SS have a characteristic profile of increased levels of ceramides, diacylglycerols DG (36:2) and DG (36:4), some TAG and a few phospholipids such as PC (32:1), PE (38:3), PE (40:6), PI (32:0) and PI (32:1). Later, in MASH patients, we found increased levels of ceramides, deoxycholic acid, a set of TAG, and some phospholipids such as PC, PE, PI and LPI; while we found decreased levels of the DG (36:0). Finally, we have reported a panel of lipid metabolites that might be used to differentiate patients with MASH from SS patients, made up of increased levels of 9-HODE some PC and PE, the LPI (16:0) and decreased levels of DG (36:0). To conclude, our investigation has suggested a lipid metabolite profile associated with MASLD and MASH. Specifically, a set of lipid metabolites seems to be discriminatory in MASH subjects compared to SS individuals. Thus, this panel of lipid metabolites could be used as a non-invasive diagnostic tool.

## Introduction

Metabolic Dysfunction-Associated Steatotic Liver Disease (MASLD) is defined by the presence of hepatic steatosis, identified through imaging or biopsy, combined with at least one metabolic alteration related to 1) body mass index (BMI) or waist circumference, 2) glucose or glycated hemoglobin A1c (HbA1c) levels, 3) blood pressure or 4) lipid profile (TAG or

**Data availability statement:** All relevant data are within the manuscript and its Supporting Information files.

**Funding:** This work was supported by Fundació URV (grant number IT20041S tu C.R.).

**Competing interests:** The authors have declared that no competing interests exist.

high-density lipoprotein cholesterol (HDL-C)) [1]. The prevalence of MASLD has increased alongside obesity and related metabolic disorders, reaching approximately 30% of the global population [2,3].

Metabolic Dysfunction-Associated Steatohepatitis (MASH), represents the severe condition of MASLD, characterized by histological features such as lobular inflammation and hepatocyte ballooning [4]. MASH carries an increased risk of progressing to fibrosis, cirrhosis or hepatocellular carcinoma [5]. Although liver biopsy is the gold standard for diagnosing and staging MASH, its invasiveness may not be suitable for all individuals [6]. Therefore, it is essential to identify non-invasive biomarkers for accurate diagnosis of MASLD and MASH subjects [7].

Given the emergence of metabolomics and lipidomics as relevant techniques for characterizing the metabolic profile in MASLD [8,9], our previous untargeted metabolomic analysis in serum samples of morbidly obese (MO) women with liver histological diagnoses focused on identifying potential metabolic factors involved in MASLD/MASH pathogenesis for use as new non-invasive biomarkers. The study specifically found lipid metabolites significantly characteristic of MASLD and MASH subjects [10].

Furthermore, previously reported metabolomic and lipidomic studies have identified metabolites involved in key metabolic pathways in MASLD pathogenesis, including alterations in amino acid metabolism and primarily in lipid metabolism [11,12]. Recent lipidomic studies have reported alterations in fatty acid, ceramide, and phospholipid metabolism, suggesting induction of MASH through mitochondrial dysfunction, oxidation and inflammation [13]. For example, ceramides have been found to play a key role in the progression of MASLD to MASH by inducing lipotoxicity [14]. Diacylglycerols and TAG are the main lipids involved in the formation of lipid droplets, which are key mediators of the lipogenesis process in MASLD [15]. Additionally, phospholipids are critical in MASLD progression due to their roles in membrane integrity, cellular signaling, and liver metabolic function. Alterations in phospholipids, such as phosphatidylcholines and phosphatidylethanolamines, can disrupt lipid transport, increase oxidative stress, and promote inflammation, contributing to hepatic dysfunction. Phospholipid imbalance also impairs mitochondrial function and reduces insulin sensitivity, accelerating the progression of MASLD to more severe stages like MASH [16]. In any case, these lipid metabolite signatures require validation in other cohorts [12,13].

In this context, our liquid chromatography coupled to a mass spectrometer (LC/MS)-based untargeted lipidomic assay aims to identify the circulating lipid metabolite profile of MASH subjects in a well-characterized cohort of MO women with hepatic histological diagnoses. We wanted to identify those lipid factors involved in MASH pathogenesis that could serve as diagnostic or predictive non-invasive biomarkers.

## Materials and methods

### Subjects

In this comprehensive investigation, we conducted an untargeted lipidomics analysis using LC/MS on serum samples obtained from a cohort of 216 women with MO (BMI ≥ 40 kg/m²) at the Hospital Universitari Sant Joan de Reus in Reus, Spain, who were slated for laparoscopic bariatric surgery.

To ensure a homogeneous subject cohort and mitigate the impact of confounding variables, particularly sex-related differences, our study exclusively focused on women. This approach is rooted in the well-established understanding that men and women exhibit significant disparities in body composition, energy balance and hormonal profiles. Moreover, prior research has also underscored sex-specific variations in lipid and glucose metabolism [17,18].

Ethical considerations were paramount throughout the study, with approval obtained from the institutional Ethics Committee board at IISPV (23c/2015). Written informed consent was diligently obtained from all participants.

This study was conducted retrospectively by accessing the patient data for the first time on June 5, 2023, and for the last time on March 4, 2024. During the data collection and analysis, the authors did not have access to the identification of the patients in the case studies because they worked on a blind and encrypted database.

Patients who had an acute illness, acute or chronic inflammatory or infective diseases, or end-stage malignant disease were excluded from this study. Menopausal women and women receiving contraceptive treatment were also excluded. In addition, women with an alcohol intake exceeding 20g per day and recurrent smokers have been excluded from this study.

## Liver histological diagnosis

Liver samples were obtained during laparoscopic bariatric surgery, in a formaldehyde solution. Subsequently, biopsies were subjected to evaluation and classification by a proficient hepatopathologist using eosin-hematoxylin staining. In this regard, women with MO and corresponding histological diagnoses were first stratified into two distinct categories: normal liver (NL, n = 44) and MASLD (n = 172) groups.

Within the MASLD cohort, further subgroups were delineated based on Kleiner *et al.*'s criteria [19]: Simple Steatosis (SS, n = 66) and MASH (n = 106). Notably, all instances of MASH were confined to grades I-II, exhibiting mild to moderate inflammation without the presence of hepatic fibrosis, aligning with stage F0 on the Ishak scoring system [20].

## Anthropometrical and biochemical variables

The anthropometric assessment encompassed the measurement of weight, height, waist-hip ratio and BMI calculation.

Specialized nurses conducted blood extraction using a BD Vacutainer® system (BD, Madrid, Spain), following an overnight fasting period just before the scheduled bariatric surgery. Venous blood samples were collected in ethylenediaminetetraacetic acid tubes, subsequently separated into plasma and serum aliquots through centrifugation (3500 rpm, 4°C, 15 min), and then stored at −80°C until processing. Biochemical analyses included parameters such as glucose, insulin, HbA1c, total cholesterol, HDL-C, low density lipoprotein-cholesterol (LDL-C), TAG, aspartate aminotransferase (AST), alanine aminotransferase (ALT), gamma-glutamyl transferase (GGT), alkaline phosphatase (ALP), lactate dehydrogenase (LDH) and ferritin levels. These analyses were conducted using a conventional automated analyzer.

## Samples preparation for the LC/MS analysis

Serum metabolites were extracted in 12 μL of isopropanol (Bio-Rad, Madrid, Spain) and vortex-mixing for 10 seconds, incubated at 4°C for 30 min and centrifuged (at 12000 rpm for 10 min at 4°C).

## LC/MS setup

LC/MS was performed with a Thermo Scientific Vanquish Horizon Ultra High Performance LC system interfaced with a Thermo Scientific Orbitrap ID-X Tribrid Mass Spectrometer (Thermo Scientific, Waltham, MA, USA).

Metabolites were separated by reverse-phase chromatography with an Acquity Ultra Performance LC C18-RP (ACQUITY UPLC BEH C18 1.7 μM, Waters). Mobile phase A was acetonitrile/water (60:40) (10mM ammonium formate), and mobile phase B was isopropanol/

acetonitrile (90:10) (10mM ammonium formate). Solvent modifiers were used to enhance ionization and to improve the LC resolution in positive and negative ionization mode. Separation was conducted under the following gradient: 0–2 min, 15–30% B; 2–2.5 min, 48% B; 2.5–11 min, 82% B; 11–11.5 min, 99% B; 11.5–12 min, 99% B; 12–12.1 min, 15% B; 12.1–15 min, 15% B.)

For MS detection, heated electrospray ionization settings were set in positive and negative ionization modes as follows: source voltage, 3.5 kV (positive), 2.8 kV (negative); ion transfer tube, 300ºC; vaporizer temperature, 300ºC; sheath gas (N2) flow rate, 50 a.u.; auxiliary gas (N2) flow rate, 10 a.u.; sweep gas (N2) flow rate, 1 a.u.; s-lens rf level, 60%; SCAN-mode; resolution, 120000 (at m/z 200); AGC target, 50%; maximum injection time, 200ms.

Quality control samples consisting of pooled samples from each condition were injected at the beginning and periodically through the workflow.

MS/MS acquisition was performed at a resolution of 15000, and the normalized collision energy of the HCD cell was stepped at 10-20-30-40%. This collision energy was tested to obtain appropriate precursor ion and product intensities. The quadrupole isolation window was 1 m/z. The maximum injection time of the C-trap was customized for each inclusion list.

Xcalibur 4.4 software (Thermo Scientific, Waltham, MA, USA) was used for LC/MS instrument control and data processing.

### LC/MS data analysis

Thermo.raw data files were transformed to.mzML using Proteowizard's MSconvert [21]. Then, mzML files were processed using RHermes [22], a computational tool that improves the selectivity and sensitivity for comprehensive metabolite profiling and identification. RHermes substitutes the conventional untargeted metabolomics workflow that detects and annotates peaks, for an inverse approach that directly interrogates raw LC/MS data points using a comprehensive list of unique molecular formulas that were retrieved from large compound-centric databases (e.g., HMDB, ChEBI, NORMAN). They are used to generate a large set of ionic formulas by combining metabolite molecular formulas with expected adduct ions depending on the polarity.

RHermes solves the limitations of peak detection by finding series of scans, named SOI (Scans Of Interest), which are defined as clusters of data points that match an ionic formula and are concentrated within a short period of time. Identification was performed by HERMES using two strategies: cosine spectral matching using an in-house DB, containing MS/MS spectra from MassBankEU, MoNA, HMBD, Riken and NIST14 databases, and using MassFrontier version 8.0 SR1 (Thermo Scientific, Waltham, MA, USA) matching against the mzCloud database. Spectral hits with high similarity scores (>0.8) were manually revised to assess correct metabolite identifications.

SOIs are qualitative elements that are suited for metabolite annotation and identification. Analytically though, only well-behaved chromatographic peaks are to be quantified, these peaks have a sharp elution profile and a high signal-to-noise ratio. In order to perform SOI quantification, SOI scan series are evaluated and partitioned into well-behaved peaks in order to extract accurate abundance values. This is performed using the qHermes R package that applies the Centwave algorithm originally found in the XCMS R package [23]. Centwave algorithm determines the baseline and boundaries of well-behaved peaks (according to a set of parameters) and assigns an abundance value using the apex (highest value) of the peak, these boundaries may be different from SOI retention time ranges.

After data quality is assured and corrections have been performed, the data is ready for statistical testing.

The complete lipidomics dataset is provided in Supporting information (S1 Table).

## Statistical analysis

All reported values are presented as either median and interquartile range (for biochemical and anthropometric variables) or mean and standard deviation (for lipidomics data), aligning with the distribution characteristics of the respective variables. Missing values in our dataset were handled using the k-nearest neighbors (KNN) imputation method. Group differences were assessed using the nonparametric Mann–Whitney test (for biochemical and anthropometric variables) or one-way ANOVA (for lipidomics data). Statistical significance was set at p-values < 0.05.

Only significant ions quantified in more than 80% of the samples underwent statistical testing for differences across experimental groups using one-way ANOVA. The obtained statistical results were subjected to adjustment through the false discovery rate (FDR) p-value correction method. Fold-changes, or ratios, were calculated by dividing the mean value of an experimental group by that of a reference group, particularly in case-control studies. These ratios, represented by log2-ratios of the means of the A/B groups, serve to filter or rank preliminary statistical results. A fold-change greater than 0 indicates higher intensity in group A compared to group B, and viceversa. To streamline the identification process, Log2FC and FDR-adjusted p-value thresholds were fine-tuned. The intention was to manage the number of metabolite entities for manual review, making the process feasible. Detailed data on the concentration of significant lipid metabolites, expressed in terms of Log2FC, mean, and standard deviation, can be found in the Supporting Information (S2-S5 Tables).

Venn diagram and Principal Component Analysis (PCA) graphs were generated with R studio software.

## Results

### Subjects

In this study, we conducted an LC/MS-based untargeted lipidomic analysis on serum samples from 216 MO women with a hepatic histological diagnosis by liver biopsy, collected during a scheduled bariatric surgery. Based on the liver histological diagnosis, subjects were categorized into three groups: NL (n = 44), SS (n = 66) and MASH (n = 106). Anthropometric and clinical parameters of the studied cohort are presented in Table 1.

All participants were women, and they were comparable in terms of age, body mass index (BMI) and waist–hip ratio. Additionally, participants did not present significant differences in accordance with cholesterol, HDL-C and LDL-C levels. In fact, it should be considered that some subjects received prolonged treatment with lipid-lowering agents. In this sense, the percentage of patients in each group treated with lipid-lowering drugs was 19.4% of NL (p = 0.657), 41.5% of SS (p = 0.482) and 23.4% of MASH (p = 0.359).

Both SS and MASH patients presented significantly higher levels of glucose, HbA1c, insulin, TAG, AST, ALT and GGT than the NL group. On the other hand, SS subjects also showed increased levels of LDH and ferritin compared to the NL group. We did not report significant differences in these parameters between SS and MASH groups.

### Lipid metabolite profile of MO women with MASLD in comparison with MO women with NL

Initially, our objective was to assess lipidomic differences between MO women with MASLD and those with NL. In this analysis, we observed elevated levels of ceramides, TAG, phosphatidylcholines (PC), phosphatidylethanolamines (PE) and phosphatidylinositols (PI), as detailed in Supporting information (S2 Table). We did not identify decreased levels of any lipid metabolite.

**Table 1. Anthropometric and biochemical parameters of the studied cohort (n = 216).**

| Variables | Normal liver (n = 44) Median (25th–75th) | Simple steatosis (n = 66) Median (25th–75th) | MASH (n = 106) Median (25th–75th) |
|---|---|---|---|
| Age (years) | 46.47 (39.27–56.16) | 47.68 (40.85–54.68) | 48.74 (40.19–56.87) |
| BMI (kg/m²) | 43.97 (41.64–49.38) | 45.89 (43.01–51.49) | 46.47 (43.31–50.55) |
| Waist-hip (m) ratio | 0.89 (0.83–0.95) | 0.93 (0.87–0.98) | 0.92 (0.87–0.98) |
| Glucose (mg/dL) | 90 (81–101) | 109.50 (92.50–133.50) *** | 105 (90–132.50) *** |
| HbA1c (%) | 5.40 (5–5.70) | 6.10 (5.47–7) *** | 5.80 (5.12–6.60) *** |
| Insulin (mUI/L) | 9.35 (5.67–13.07) | 19 (11.14–33) *** | 16.34 (11.50–25.14) *** |
| TAG (mg/dL) | 103.50 (77–135.50) | 151 (116–197) *** | 146 (116–207) *** |
| Cholesterol (mg/dL) | 164 (140.50–200.25) | 164.80 (144–192) | 164.65 (146.50–186.75) |
| HDL-C (mg/dL) | 39.50 (32.40–50.30) | 36.75 (32–46) | 38 (32.35–43) |
| LDL-C (mg/dL) | 103.05 (79.32–127.35) | 90.90 (76.25–114) | 95 (76.17–116) |
| AST (UI/L) | 22 (19–39) | 35.35 (24.25–52.50) *** | 35 (25–54.25) *** |
| ALT (UI/L) | 23 (16–44.50) | 35 (29–49) *** | 34 (25–58) *** |
| GGT (UI/L) | 17 (12–26.25) | 26.25 (18.75–45.75) *** | 22.10 (15–50.75) * |
| ALP (Ul/L) | 65 (52.50–76.50) | 68 (54.50–76) | 68 (58.80–78) |
| LDH (Ul/L) | 388 (340–423.50) | 427.50 (351.75–476.75) * | 396.50 (344.25–481) |
| Ferritin (ng/mL) | 36 (21.50–76.75) | 75.14 (33.42–185.49) * | 54 (27–119.30) |

Data are expressed as the median and interquartile range. Differences between NL and SS or MASH were considered significant when (*) p-value < 0.05, (**) p-value < 0.1 or (***) p-value < 0.005 using the Mann–Whitney test. MASH, metabolic dysfunction-associated steatohepatitis; BMI, body mass index; HbA1c, glycosylated hemoglobin A1c; TAG, triacylglycerols; HDL-C, high-density lipoprotein-cholesterol; LDL-C, low-density lipoprotein-cholesterol; AST, aspartate aminotransferase; ALT, alanine aminotransferase; GGT, gamma-glutamyl transferase; ALP, alkaline phosphatase; LDH, lactate dehydrogenase.

Subsequently, our focus shifted to examining the lipid metabolite profile of the two stages of MASLD (SS and MASH) independently in comparison to patients with NL histology.

### Lipid metabolite profile MO women with SS compared to those with NL

On one hand, when assessing the lipidome of MO women with SS in comparison to those with NL, we once again observed heightened levels of lipid species, including ceramides, diacylglycerols, TAG, PC, PE and PI, as outlined in Supporting information (S3 Table). Again, we did not identify decreased levels of any lipid metabolite.

### Lipid metabolite profile in MO women with MASH compared to those with NL

Later, our attention turned to examining the lipidomic profile of MASH subjects in comparison to those with NL histology. In this context, our findings revealed that individuals with MASH exhibited elevated levels of ceramides, the bile acid deoxycholic acid, TAG, PC, PE, lyso-PI (LPI) and PI. Additionally, decreased levels of the diacylglycerol DG (36:0) were observed in the MASH group compared to the NL participants, as detailed in Supporting information (S4 Table).

### Lipid metabolite profile in MO women with MASH compared to those with SS

In our final analysis, we compared MO patients with MASH to those with SS. In this comparison, we observed heightened levels of the fatty acid 9-hydroxy-octadecadenoic acid (9-HODE), PC, PE and the LPI (16:0) in MASH women, while decreased levels of the DG

(36:0) were reported in MASH compared to SS. These findings are detailed in Supporting information (S5 Table).

## Comparative analysis of significant variables between groups

Since we have found a common pattern of lipid metabolites in the comparisons between groups but have also observed that some metabolites are distinctive to the studied groups, we wanted to conduct a PCA to illustrate the distribution of these lipid metabolites among the groups (Fig 1), although PCA reflects general variance. In this case, we can see that the metabolites do not show any separation between the NL and SS groups and largely overlap with the MASH group, although there are some metabolites that are present only in the latter group.

Then, we have included a Venn diagram to illustrate the distribution and overlap of significant lipid metabolites across the different comparisons (Fig 2). The diagram shows the variables that are exclusive to each comparison, as well as those shared between multiple comparisons. The list of the lipid metabolites present in this graph are detailed in the Supporting information (S6 Table). In the Venn diagram, we can observe that there are no significant lipid metabolites between MASLD and NL, but there are 11 between SS and NL, 10 between MASH and NL, and 6 between MASH and SS. There is only one lipid metabolite common between MASLD and NL and SS vs NL, 2 both in SS and MASH vs NL, and one between MASH vs NL and MASH vs SS. On the other hand, there are 4 significantly shared metabolites in MASLD and MASH vs NL, and 14 between MASLD vs NL, SS vs NL, and MASH vs NL. For the rest of

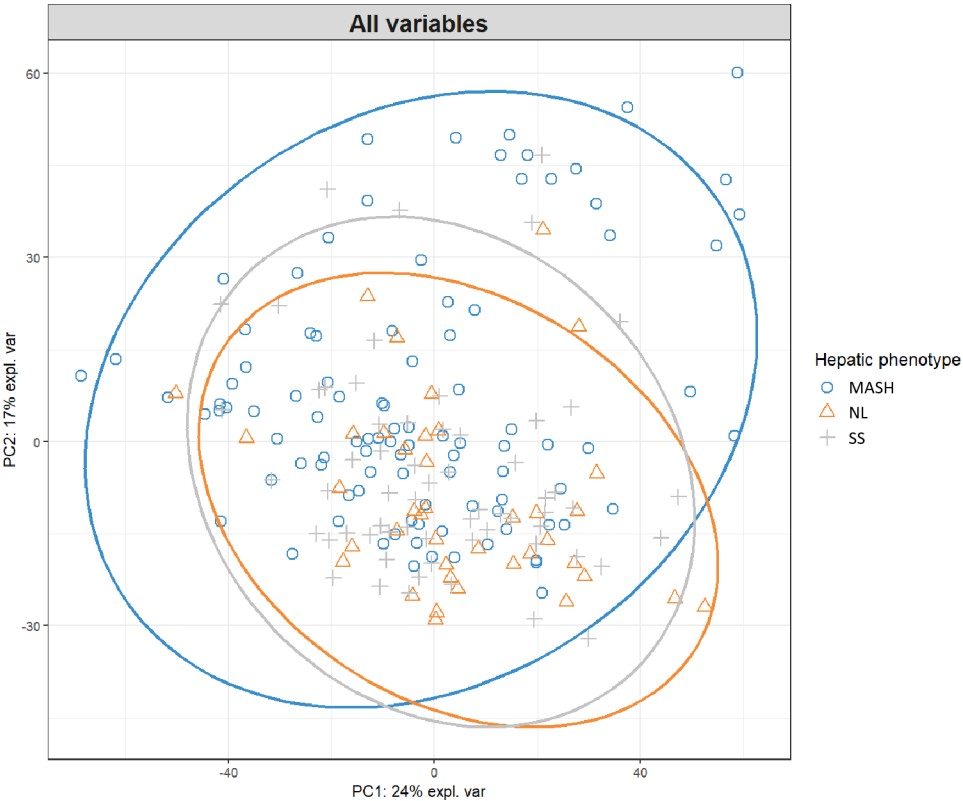

**Fig 1. Principal Component Analysis (PCA) to illustrate the distribution of the significant lipid metabolites among the groups (NL, SS and MASH).** MASH, metabolic-dysfunction associated steatohepatitis; NL, normal liver; SS, simple steatosis.

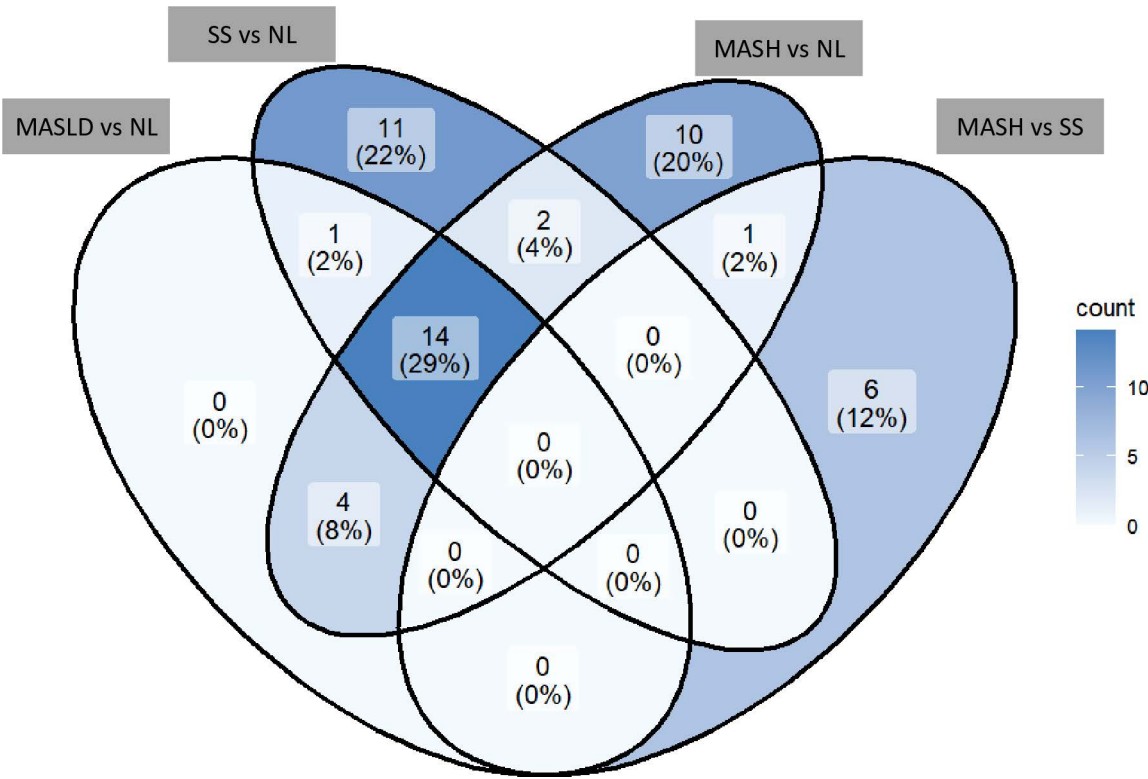

**Fig 2. Venn diagram to illustrate the distribution and overlap of significant lipid metabolites across the different comparisons.** MASLD, metabolic-dysfunction associated steatotic liver disease; NL, normal liver; SS, simple steatosis; MASH, metabolic-dysfunction associated steatohepatitis.

the combinations, there are no significant lipid metabolites for the Venn diagram analysis. In this analysis, only the metabolites significantly increased between groups were evaluated, as the only decreased metabolite was DG (36:0). This analysis highlights that, although there are 14 common lipid metabolites across most comparisons, the comparison of MASH vs SS is an exception, with 6 lipid metabolites characteristic of it.

## Discussion

This study presents comparisons between the characteristic lipid metabolite profiles of each histopathological stage in women with MO. This work, reporting an analysis using LC/MS of untargeted lipidomics in a cohort of 216 participants, has found lipid metabolites mostly with increased concentrations in different stages of MASLD, allowing the definition of which of these metabolites are differential between stages of the disease and therefore potential biomarkers.

Firstly, in this study, we have identified a profile of lipid metabolites that are increased in women with MASLD associated with MO, composed of ceramides, TAG, and some phospholipids such as phosphatidylcholines, PE and PI. On the other hand, we have identified that those patients with SS have a characteristic profile of increased levels of ceramides, diacylglycerols DG (36:2) and DG (36:4), some TAG, PC (32:1), PE (38:3), PE (40:6), PI (32:0) and PI (32:1). Then, in those patients with MASH, we found increased levels of ceramides, the bile acid deoxycholic acid, a set of TAG, and some phospholipids such as PC, PE, PI and LPI; while we found decreased levels of the DG (36:0). Finally, we have reported a panel of

lipid metabolites that are discriminatory and specific of patients with MASH compared to SS patients, made up of increased levels of the oxylipin 9-HODE some PC and PE, the LPI (16:0) and decreased levels of DG (36:0).

In a previous metabolomics study, we also found increased levels of PC (32:1) in MASLD patients compared to NL, such as in the current study, but we also found increased levels of PC (32:2) in MASLD compared to NL and increased levels of DG (36:2) and PE (40:6). In the current study, we have found these trends in MASH compared to NL and in SS compared to NL, respectively [10]. Then, we agree again with our previous study because we have reported increased PE (40:6) levels in SS compared to NL and increased levels of PC (32:1), PC (32:2) and PI (32:1) in MASH compared to NL [10].

In this sense, a recent lipidomics study by Núñez-Sánchez *et al.* revealed that alterations in the lipidome were present in liver samples of MASLD subjects but not in serum samples [24]. Conversely, Chen *et al.* suggested that there are different alterations in serum metabolome and lipidome at the onset and progression of MASLD in a cohort of obese Chinese [25]. Their results differ from ours since they found mostly decreased levels of phospholipids such as PC and PI in the patients as their histopathological stage progressed. However, we agree with them reporting increases in TAG as the stage of liver disease progressed [25].

With this study, we wanted to find a differential lipid metabolite panel between patients with MASH and those with SS. However, we have not found concordance with respect to the lipidomics profile differences between MASH and SS with our previous study. In this regard, Puri *et al.* in 2009 reported decreased levels of monounsaturated fatty acids and increased levels of the oxylipines 5-, 8-, 11- and 15-hydroxy-eicosatetraenoic acid (HETE) in plasma samples from 50 MASH subjects compared to 25 SS subjects [26]. Furthermore, Barr *et al.* in 2012 performed an analysis of lipidomics by UPLC-MS in plasma samples, finding increased levels of 5-, 9-, 11-, 12- and 15- HETE in a group of 131 NASH subjects from a cohort of 246 MASLD individuals [27]. In this case, we have also reported increased levels of an oxylipin in MASH compared to SS, but it was the 9-HODE. These oxylipins have been suggested as biomarkers of MASLD progression [28]. The involvement of HETE and HODE in the pathogenesis of MASLD was validated by a noteworthy decline in these examined eicosanoids and a parallel reduction in fatty liver [29]. On the other hand, Gorden *et al.* in 2015, performing the same analysis, reported decreased levels of LPE and sphingomyelins, and increased levels of PE and ceramides in 20 MASH subjects compared to 17 individuals with SS [30]. In our study, we also observed increased levels of PE, specifically PE (O-36:3) and PE (P-34:3). Studies have indicated that maintaining balance in membrane phospholipids is crucial for the development and advancement of metabolic diseases. Grapentine *et al.* demonstrated for the first time that an unfavorable metabolic energy state, caused by decreased synthesis of membrane lipids, which prompts excessive fatty acid production to handle unused intermediates, could trigger all features of MASH, such as steatosis and inflammation [31]. On the other hand, Wu *et al.* reported the depletion of PE in MASH, indicating a potential role in the disease [32]. However, more research is needed to fully understand the relationship between PE and MASH. Then, Velenossi *et al.* in 2022 found increased diacylglycerol levels in NASH subjects [33]; while we have found decreased levels of the DG (36:0). Similarly, Okour *et al.* also showed that diacylglycerol levels, specifically DG (36:0), are decreased in subjects with MASH [34]. This decrease in diacylglycerol levels could be influenced by factors such as the use of statins in MASH patients with dyslipidemia [35].

In this regard, and supported by our analyses, we suggest that there is a different profile of lipid metabolites between patients with MASH compared to those with SS. This, in addition to the comparative analysis between groups, has been corroborated in both the PCA and the Venn diagram. Therefore, these different metabolites would allow us to discriminate the most

important group to differentiate, which is MASH, from those patients who only suffer from SS due to the severity of the condition.

As a limitation of this work, we can highlight that we have evaluated a cohort of fertile women with MO, but we wanted to focus on a homogeneous cohort to avoid bias regarding sex, age and degree of obesity. Furthermore, given that this analysis was performed in a considerable cohort of subjects with promising results, it is necessary to perform targeted omics analyses in validation cohorts to confirm these results. We could also mention that no exclusive lipid metabolite profile was shown in the PCA, so in this regard, the discriminative potential of the model is limited, although PCA reflects general variance. In addition, as a technical limitation, the LC/MS technique is susceptible to changes in its optimal conditions, such as temperature and atmospheric pressure changes, solvent impurities or hardware inconsistencies. MS might detune during an experiment or lose sensitivity during an experiment. How we have amended these biases was previously defined elsewhere [36].

To sum up, this study highlights the potential of lipid metabolite profiling to distinguish between MASH and SS, with interesting results from a homogeneous cohort of fertile women with morbid obesity, minimizing confounding factors like sex and age. While the PCA and Venn diagram analyses support the findings, limitations include the restricted generalizability to other populations, the absence of an exclusive lipid metabolite profile, and the inherent technical variability of LC/MS techniques. Further validation in diverse cohorts and refined omics analyses are needed to confirm these findings and enhance the model's discriminative power.

## Conclusion

To conclude, in this LC/MS-based untargeted lipidomics study, meticulously conducted on a homogeneous cohort of fertile women with MO, our investigation has reported a lipid metabolite profile associated with MASLD and MASH. Specifically, we found a set of increased lipid metabolites such as 9-HODE oxylipin, some PC, PE and the LPI (16:0), and decreased levels of the DG (36:0). This set of lipid metabolites seems to be discriminatory in MASH subjects compared to SS individuals. Thus, this panel of lipid metabolites could be used as a non-invasive diagnostic tool.

## Supporting Information

**S1 Table. Complete lipidomics dataset.** The raw data obtained from the untargeted lipidomics study of the analyzed serum samples are presented in the tables according to the negative or positive ionization of the process.
(XLSX)

**S2 Table. Log2 fold change (FC), mean and standard deviation (SD) of each lipid specie from the normal liver (NL) and metabolic dysfunction-associated steatotic liver disease (MASLD) groups, and adjusted p-values.** Data from significant lipid metabolites concentration expressed in the Log2FC, mean and SD. One-way ANOVA test was used to identify significant differences (adjusted p-value < 0.05 was considered significant).
(XLSX)

**S3 Table. Log2 fold change (FC), mean and standard deviation (SD) of each lipid specie from the normal liver (NL) and simple steatosis (SS) groups, and adjusted p-values.** Data from significant lipid metabolites concentration expressed in the Log2FC, mean and SD. One-way ANOVA test was used to identify significant differences (adjusted p-value < 0.05 was considered significant).
(XLSX)

**S4 Table. Log2 fold change (FC), mean and standard deviation (SD) of each lipid specie from the normal liver (NL) and metabolic dysfunction-associated steatohepatitis (MASH) groups, and adjusted p-values.** Data from significant lipid metabolites concentration expressed in the Log2FC, mean and SD. One-way ANOVA test was used to identify significant differences (adjusted p-value < 0.05 was considered significant). (XLSX)

**S5 Table. Log2 fold change (FC), mean and standard deviation (SD) of each lipid specie from the simple steatosis (SS) and metabolic dysfunction-associated steatohepatitis (MASH) groups, and adjusted p-values.** Data from significant lipid metabolites concentration expressed in the Log2FC, mean and SD. One-way ANOVA test was used to identify significant differences (adjusted p-value < 0.05 was considered significant). (XLSX)

**S6 Table. Log2 fold change (FC) of the overlaps of lipid metabolites between the study groups and those significant metabolites that appear exclusively in a single group (unique) represented in the Venn diagram.** Data from significant lipid metabolites concentration expressed in the Log2FC. One-way ANOVA test was used to identify significant differences (adjusted p-value < 0.05 was considered significant). (XLSX)

## Acknowledgments

Thanks to the Fundació URV for its administrative collaboration.

## Author contributions

**Conceptualization:** Cristóbal Richart.

**Data curation:** Laia Bertran, Jordi Capellades.

**Formal analysis:** Laia Bertran, Cristóbal Richart.

**Funding acquisition:** Cristóbal Richart.

**Investigation:** Laia Bertran, Cristóbal Richart.

**Methodology:** Jordi Capellades, Sonia Abelló.

**Project administration:** Cristóbal Richart.

**Resources:** Sonia Abelló.

**Software:** Laia Bertran, Jordi Capellades.

**Supervision:** Cristóbal Richart.

**Validation:** Cristóbal Richart.

**Visualization:** Cristóbal Richart.

**Writing – original draft:** Laia Bertran.

**Writing – review & editing:** Cristóbal Richart.

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
